# Variability of the Indian Ocean Dipole post-2100 reverses to a reduction despite persistent global warming

Guojian Wang [1], Wenju Cai [1,2,3,4,5] & Agus Santoso [1,6,7]

Previous examination of the Indian Ocean Dipole (IOD) response to greenhouse warming shows increased variability in the eastern pole but decreased variability in the western pole before 2100. The opposing response is due to a shallowing equatorial thermocline promoting sea surface temperature (SST) variability in the east, but a more stable atmosphere decreasing variability in equatorial zonal winds that weakens SST variability in the west. Post-2100, how the IOD may change remains unknown. Here we show that IOD variability weakens post-2100 in majority of models under a long-term high emission scenario to 2300. Post-2100, the atmosphere stability increases further and persistent ocean warming arrests or even reverses the eastern Indian Ocean shallowing thermocline. These changes conspire to drive decreased variability in both poles, reducing amplitude of moderate, strong and early-maturing positive IOD events. Our result highlights a nonlinear response of the IOD to long-term greenhouse warming under the high emission scenario.

The Indian Ocean Dipole (IOD) refers to a sea surface temperature (SST) seesaw pattern between the western equatorial Indian Ocean (WEIO) and the eastern equatorial Indian Ocean (EEIO), and is a prominent mode of interannual climate variability in the tropical Indian Ocean[1–3]. An IOD usually develops in austral winter (June–July–August, JJA) and matures into austral spring (September–October–November, SON). During a positive IOD (pIOD), easterly wind anomalies along the tropical Indian Ocean pile up warm water to the west and drive an increased equatorial oceanic upwelling and a shallowed thermocline, cooling the SST in the east. The intensified west-minus-east SST gradient, in turn, reinforces the easterly wind anomalies, in a positive feedback involving wind, thermocline and SST[1]. The associated convective heating anomalies trigger tropical baroclinic response and extratropical barotropic Rossby wave trains that generate weather and climate extremes on a global scale[4,5]. For example, the 1997 pIOD caused devastating floods in eastern Africa but catastrophic wildfires

in Indonesia[6], and such disasters occurred again during the 2019 pIOD leading to "Black Summer" bushfires in Australia[7]. Because of these severe impacts, understanding how the IOD may change under greenhouse warming is one of the most important climate issues.

Projection based on the comparison between the 20th century and the 21st century shows a weak inter-model consensus on change in variability in the classic Dipole Mode Index (DMI)[2,8,9], defined as the west-minus-east difference in SST anomalies between the WEIO (50°E–70°E, 10°S–10°N) and the EEIO (90°E–110°E, 10°S–0°)[1]. The weak inter-model consensus on changes in DMI variability is due to offsetting effects of mean state changes in the atmosphere and the ocean[2,8]. On the one hand, the troposphere is projected to warm faster than the surface, leading to a more stable atmosphere with reduced surface zonal wind variability, which weakens wind-associated positive feedbacks, particularly zonal advection that dominates WEIO SST variability[2,8,10]. On the other hand, the WEIO is projected to warm faster

[1]CSIRO Environment, Hobart, TAS, Australia. [2]Frontier Science Center for Deep Ocean Multispheres and Earth System (FDOMES) and Physical Oceanography Laboratory, Ocean University of China, Qingdao, China. [3]Laoshan Laboratory, Qingdao, China. [4]State Key Laboratory of Marine Environmental Science & College of Ocean and Earth Sciences, Xiamen University, Xiamen, China. [5]State Key Laboratory of Loess and Quaternary Geology, Institute of Earth Environment, Chinese Academy of Sciences, Xi'an, China. [6]International CLIVAR Project Office, Ocean University of China, Qingdao, China. [7]Australian Research Council (ARC) Center of Excellence for Climate Extremes and Climate Change Research Centre, University of New South Wales, Sydney, NSW, Australia. e-mail: Wenju.Cai@csiro.au; a.santoso@unsw.edu.au

than the EEIO during the 21st century in association with enhanced oceanic upwelling and shoaling thermocline in the EEIO. This promotes nonlinear advection and the SST response to the thermocline, positive feedbacks that are favourable to a subsurface-induced cooling on the SST over the EEIO[2,8,9,11]. In association, there is a projected decrease in amplitude of moderate-pIOD dominated by WEIO SST variability, but a tendency for increased amplitude and frequency of strong-pIOD dominated by EEIO[9]. However, how the IOD may respond to increasing greenhouse warming after the 21st century is unclear. Below, we show that IOD SST variability weakens from 2100 to 2300 despite a continuous increase in global mean temperature.

## Post-2100 reduction in variability of both IOD poles

In two generations of models that participate in Coupled Model Intercomparison Project phase 5 (CMIP5)[12] and phase 6 (CMIP6)[13], nine models from CMIP5 and eight models from CMIP6 forced under historical and a business-as-usual emission scenario project to the end of the 23rd century (see 'CMIP data and processing' in 'Methods' and Supplementary Table 1). In these scenarios, atmospheric $CO_2$ concentration continues to increase to roughly 2000 ppm by 2250[14], generating continuous warming in the global mean temperature (red curve in Fig. 1a).

We calculated the DMI from 1860 to 2300 for each model. To remove the influence from variability on decadal timescales and longer, monthly SST anomalies for each year are referenced to the monthly climatology of a 10-year running average (see 'CMIP data and processing' in 'Methods'). The time evolution of the DMI variability is then assessed in a 100-year sliding window focusing on SON average when an IOD matures, with the resultant amplitude denoted on the central year such that value at 2050, for instance, represents the IOD amplitude over the 21st century. Those models simulate reasonable IOD pattern and phase locking compared to the observed (Supplementary Fig. 1).

In contrast to the mild change from the 20th to the 21st century, DMI variability decreases after 2100 and such a decrease continues into the end of the 23rd century (Fig. 1a). Comparing among the 21st, 22nd, and 23rd centuries, 16 out of 17 models project a continuous weakening in DMI variability and the difference in their multi-model ensemble mean is statistically significant above the 95% confidence level (Fig. 1b). Beyond 2050, time evolution of the multi-model ensemble mean SST variability over the western and eastern pole of the IOD resembles that of the DMI, both projected to decrease towards 2300 (Fig. 1a, c, e). However, WEIO SST variability peaks during the 20th century (Fig. 1c, d), whereas EEIO SST variability peaks during the

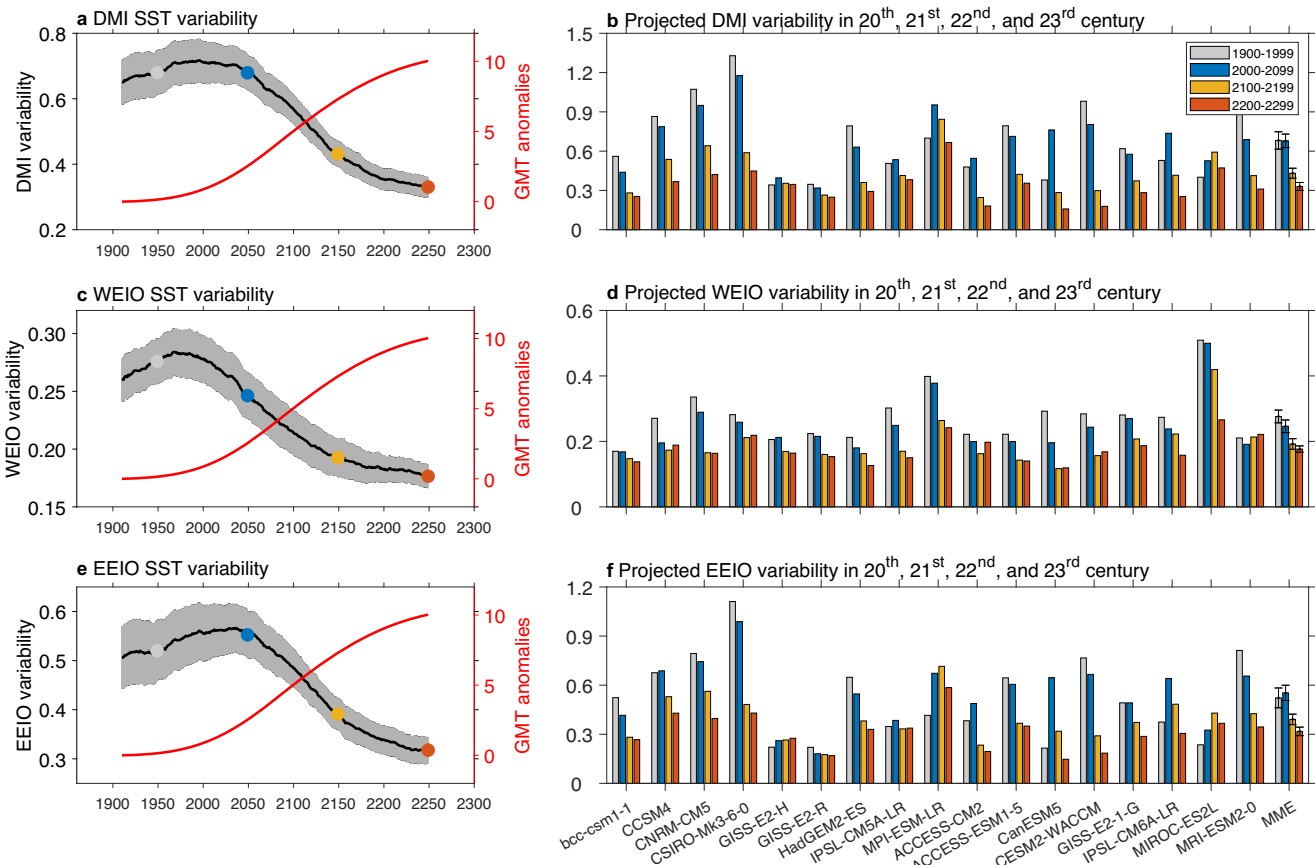

**Fig. 1 | Time evolution of Indian Ocean Dipole (IOD) variability in a 100-year sliding window from 1860 to 2300. a** September, October, and November (SON) SST variability of the Dipole Mode Index (DMI), defined as the difference in sea surface temperature (SST) anomalies between the Western Equatorial Indian Ocean (WEIO; 50°E–70°E, 10°S–10°N) and the Eastern Equatorial Indian Ocean (EEIO; 90°E–110°E, 10°S–0°)[1]. The black curve indicates the multi-model ensemble mean and the grey shading indicates the one-standard-deviation range based on a bootstrap test[18] (see 'Statistical significance test' in 'Methods'). The red curve shows the multi-model ensemble mean of global mean temperature (GMT) in a 100-year

running average, and then subtracting the mean GMT of the first 100-year window. **b** DMI variability over the 20th, the 21st, the 22nd, and the 23rd centuries for each model corresponding to values indicated, respectively, by the grey, blue, yellow, and orange dots in (**a**). The error bar with the multi-model ensemble mean is one-standard-deviation range based on bootstrap. **c–f** The same as (**a, b**), but for WEIO SST variability and EEIO SST variability, respectively. Despite a continuous global warming into 2300, DMI variability shows a reduction after 2100. Before 2100, WEIO SST variability decreases but EEIO SST variability increases; SST variability in both poles decreases thereafter.

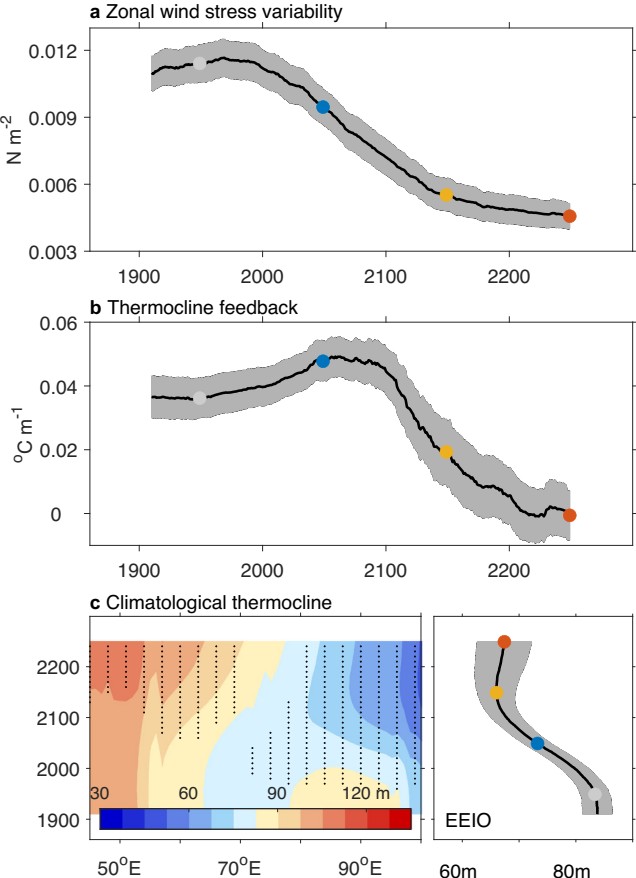

**Fig. 2 | Mechanism of Indian Ocean Dipole (IOD) time evolution under long-term greenhouse warming. a** Time evolution in September, October, and November (SON) zonal wind variability towards 2300 over the central tropical Indian Ocean (5°S–5°N, 60°E–100°E). **b** The same as (**a**), but for the thermocline feedback over the Eastern Equatorial Indian Ocean (EEIO). It is calculated as the regression coefficient of sea surface temperature (SST) onto the thermocline anomalies for each of the 100-year window. **c** Time evolution of annual mean thermocline depth (m) along the equatorial Indian Ocean averaged over 5°S–5°N in a 100-year sliding window. The dotted area indicates regions where the climatological thermocline is statistically different from the mean thermocline over the 1860–1900 above the 90% confidence level based on a Student's *t* test. The time evolution of EEIO annual mean thermocline depth is plotted on the right panel. SON SST variability in WEIO and EEIO is dominated by zonal wind variability and by the thermocline feedback, respectively. After the 21st century, both zonal wind variability and the thermocline feedback weakens, leading to reduced SST variability in both poles of the IOD.

21st century (Fig. 1e, f). The earlier weakening in WEIO variability is supported by a strong inter-model consensus, with 16 out of 17 models showing a persistent weakening (Fig. 1d). The weakening offsets an increase in EEIO SST variability, leading to the lack of an inter-model consensus in DMI variability before 2100. After the 21st century, WEIO and EEIO variability decrease concurrently, resulting in decreased DMI variability towards 2300.

**Thermocline feedback post-2100 weakens**
We calculated anomalies in zonal wind stress and thermocline in a similar manner as for SST anomalies. Over the 20th and the 21st century, time evolution of zonal wind variability and the thermocline feedback show a consistent change with that of the WEIO and the EEIO SST variability, i.e., peaking in the 20th century (Fig. 2a) and the 21st century (Fig. 2b), respectively, before a continuous reduction after the

21st century. The majority of models project a reduction in zonal wind variability but a strengthening in the thermocline feedback in the 21st century compared to the 20th (grey vs blue bars in Supplementary Fig. 2): models with a greater weakening in zonal wind variability project a greater reduction in DMI variability (Supplementary Fig. 3a), while models with a greater increase in the thermocline feedback project stronger DMI variability (Supplementary Fig. 3d), confirming their initial offsetting impacts on the projected DMI variability changes[2,8]. The weakening in DMI variability could have started from the 20th century due to the weakening in zonal wind variability, but have been delayed by the strengthening in the thermocline feedback due to the shoaling thermocline over the EEIO[8].

The changes in zonal wind variability and thermocline feedback after the 21st century could be explained by changes in the atmospheric and oceanic mean state. As the troposphere continues to warm faster than the surface (Supplementary Fig. 4a), a weakening in zonal wind variability persists (Supplementary Fig. 2a), contributing to decreasing DMI variability (Supplementary Fig. 3b, c). Further, thermocline feedback weakens, or no longer operates after 2100 as indicated by a negative value after the 21st century in some models (Supplementary Fig. 2b). We found that the mean state of EEIO thermocline no longer shallows or even deepens after the 21st century (Fig. 2c). Before 2100, equatorial easterly trends in association with the weakening Walker Circulation push warm water towards the west (Supplementary Fig. 4b). After 2100, warm water piled up in the west extends eastward (Supplementary Fig. 4b). As a result, the eastern warming increases (Supplementary Fig. 4c), eventually overwhelming and arresting the shoaling EEIO thermocline, weakening the thermocline feedback in the east (Fig. 2b). There is a coherent time evolution in the climatological EEIO thermocline, west-minus-east SST gradient and zonal wind stress (Supplementary Fig. 5). Thus, after 2100 the continuous weakening in the thermocline feedback conspires with the persistent weakening in zonal wind variability, driving weakened DMI SST variability in the 22nd and 23rd centuries.

In association with decreased IOD variability, there is a corresponding change in SST-cloud-radiation feedback, a major negative feedback during IOD growth and maturity phase[2,3]. We calculate the response of surface shortwave radiation (SWR) to SST anomalies in the EEIO, in which the SWR anomalies are constructed following the same procedure as applied to the DMI. There is a continuous weakening in the amplitude of the SWR response, indicating that the IOD thermal damping is weakening (Supplementary Fig. 6). Thus, the weakening of the positive Bjerknes feedback is the primary cause for the decreasing IOD variability, which would otherwise have increased because of the weakening in the damping.

The IOD interacts with El Niño-Southern Oscillation (ENSO) event[2,3]. To assess whether decreased DMI variability is in any way linked to changes in ENSO, we construct an inter-model relationship between the projected changes in DMI variability and in Niño3.4 variability. There is a significant correlation coefficient between projected changes from the 20th to the 21st century, implying that ENSO could contribute to the change in IOD variability (Supplementary Fig. 7a). However, the correlation disappears after the 21st century when DMI variability decreases, ruling out ENSO as a cause for the decrease in IOD variability, although ENSO variability decreases in the majority of models (Supplementary Fig. 7b, c).

**Long-term reduction in strong, moderate, and early-pIOD variability**
The IOD features strong diversity in event intensity and in spatial anomaly patterns[9,15]. To examine the response of "strong-pIOD" and "moderate-pIOD", which are dominated by cool SST anomalies over the EEIO and warm anomalies over the WEIO, respectively, we applied

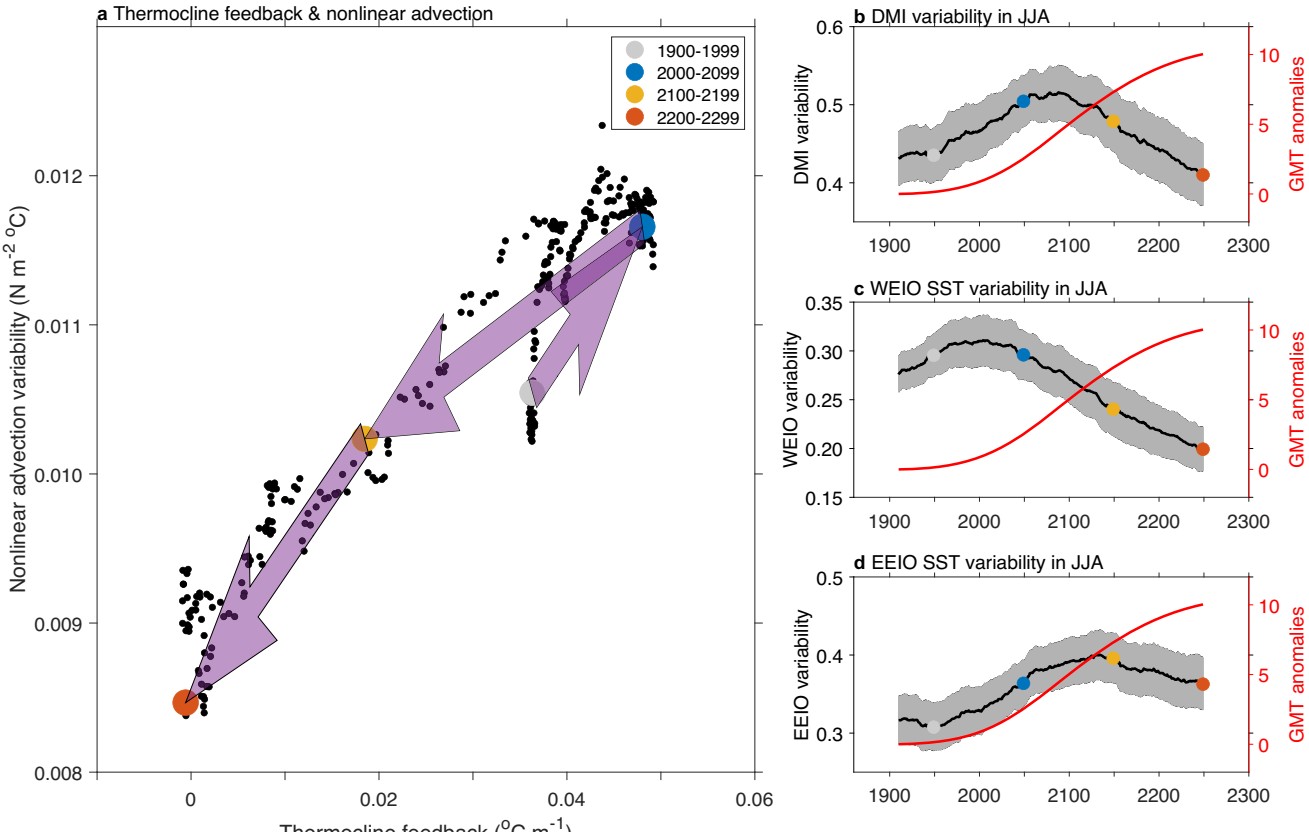

**Fig. 3 | Mechanism of strong positive Indian Ocean Dipole (pIOD) time evolution and time evolution for early-pIOD under long-term greenhouse warming.**
**a** The relationship of multi-model ensemble mean of the time evolution in the thermocline feedback and the nonlinear advection feedback. The arrows indicate magnitude change in the thermocline feedback and the nonlinear advection from the 20th to the 21st, from the 21st to the 22nd, and from the 22nd to the 23rd century. **b**–**d** The same as Fig. 1a, c, e, but for sea surface temperature (SST)

variability of the Dipole Mode Index (DMI), Western Equatorial Indian Ocean (WEIO), and Eastern Equatorial Indian Ocean (EEIO), respectively, over the June, July, and August (JJA) season. The changes in nonlinear advection and thermocline feedback which are closely linked, translates to the increase-then-decrease time evolution in strong-pIOD. In contrast to the IOD in SON, the DMI in JJA shows a distinct increase-then-decrease time evolution, governed by the much stronger thermocline feedback in JJA over the 21st century.

an EOF analysis on the SON SST anomalies over the tropical Indian Ocean[9] (see 'Strong-pIOD and moderate-pIOD' in 'Methods'). The index for strong-pIOD (S-index) and moderate-pIOD (M-index) events can be calculated as $(PC1 + PC2)/\sqrt{2}$ and $(PC1 - PC2)/\sqrt{2}$, respectively. The time evolution of moderate-pIOD and strong-pIOD SST variability correspondingly resembles that of the WEIO and EEIO SST variability (Supplementary Fig. 8a, c). The persistent decrease in M-index variability from the 20th century (Supplementary Fig. 8a, b) is due to a weakening in zonal wind variability[9]. The strong-pIOD is governed by an easier movement of atmospheric convection to the west that is conducive to equatorial nonlinear advection (see 'Nonlinear advection' in 'Methods') and the thermocline feedback[9]; the two positive feedbacks are closely linked (Fig. 3a) as nonlinear advection feeds onto the thermocline feedback. Both are projected to increase during the 21st century[6,9] but decrease thereafter (see arrows in Fig. 3a), leading to the change in strong-pIOD variability with an increase in the 21st century but a decrease thereafter (Supplementary Fig. 8c, d).

The IOD also features diversity in event peak season[16,17]. To examine "early-pIOD" that matures in JJA, we use JJA DMI variability, which is projected to increase from the 20th to the 21st century (Fig. 3b), consistent with the projected increase in the early-pIOD events during the 21st century[16,17]. However, a weakening in zonal wind variability (Supplementary Fig. 9a, b), and a decrease after the 22nd century in the thermocline feedback (Supplementary Fig. 9c, d), likewise dominate the time evolution in the WEIO and EEIO SST variability

(Fig. 3c, d), leading to an eventual reduction in JJA DMI variability towards the end of the 23rd century (Fig. 3b).

### Sensitivity to emission scenarios
The decrease in IOD variability after the 21st century arises from the persistent mean state change in both atmosphere and ocean under the business-as-usual emission scenario. A question arises as to how our result is sensitive to emission scenario. To this end, we repeat the analysis but using available models forced under a low $CO_2$ emission scenario, that is, RCP2.6 for CMIP5 and SSP1-2.6 for CMIP6 (Supplementary Table 1). In contrast to the business-as-usual scenario, in the low emission scenario there is no further increase in atmospheric stability after about 2100 (Supplementary Fig. 10a vs Supplementary Fig. 4a). Similarly, the warming over the entire tropical Indian Ocean stabilises after the 21st century with little zonal contrast in oceanic warming rates (Supplementary Fig. 10b, c). Thus, the stabilised warming in the lower troposphere and in the tropical ocean, together with stabilised zonal wind variability and thermocline feedback (Supplementary Fig. 11), leads to stabilised SST variability across the tropical Indian Ocean (Supplementary Fig. 12), in contrast to the persistent reduction in IOD variability under the high-emission scenario.

### Discussion
Changes in austral winter or spring DMI SST variability from the 20th to the 21st century result from the offsetting effects of reduced zonal

wind variability but an intensified thermocline feedback. However, persistent warming after the 21st century not only continues to warm the lower troposphere but also causes the initially shallowing of the equatorial eastern Indian Ocean thermocline and the initially faster warming in the western Indian Ocean than the east to stop or even reverse. Thermocline feedback, which initially boosts IOD variability, weakens substantially after 2100, conspiring with the persistent long-term decrease in zonal wind variability to weaken IOD variability in both seasons. Our finding highlights a strongly nonlinear behaviour pattern of the IOD response to long-term greenhouse warming under the high-emission scenario.

## Methods
### CMIP data and processing
To assess the time evolution in the Indian Ocean Dipole in response to long-term greenhouse warming beyond the 21st century, we used nine models from CMIP5[12] and eight models from CMIP6[13] forced under historical forcing (1860–2005 for CMIP5 and 1860–2014 for CMIP6) and the business-as-usual emission scenario (2006–2300 for CMIP5 RCP8.5 and 2015–2300 for CMIP6 SSP5-8.5) (Supplementary Table 1). Monthly data of surface temperature, SST zonal wind stress, ocean temperature, air temperature, and surface shortwave radiation (SWR) are utilised, to calculate global mean temperature, IOD variability, zonal wind variability, thermocline depth defined as the depth for maximum vertical gradient, atmospheric stability, and SWR response to SST, respectively. Before data analysis, the horizontal grids of each model are regridded to $1° \times 1°$. The scenario includes an increase in $CO_2$ concentration, before a stabilisation around 2250 at roughly 2000 ppm, or more than 7 times the pre-industrial $CO_2$ concentrations[14].

To exclude the possible influence from decadal variability and beyond, we firstly calculate monthly SST anomalies for each year by subtracting the monthly climatology over a 10-year window that centres the year. We then calculate its variability in a 100-yr sliding window focusing on SON season. In this way, DMI SST variability for each century in each model is represented by SST variability centered at the year of mid-century. The same procedure applies to zonal wind variability and thermocline feedback where anomalies are used. The zonal wind variability is calculated over the central tropical Indian Ocean (5°S–5°N, 60°E–100°E) and the thermocline feedback is calculated as the regression coefficient of SST onto thermocline anomalies over the EEIO. Moreover, analysis is based on JJA season for "early-pIOD" and annual mean for the climatological mean state.

To evaluate the sensitivity of IOD variability changes to different future warming pathways, we also use data forced under the low $CO_2$ emission scenario, i.e., CMIP5 RCP2.6 and CMIP6 SSP1-2.6 (Supplementary Table 1).

### Strong-pIOD and moderate-pIOD
To examine the time evolution of strong-pIOD and moderate-pIOD, we apply an EOF analysis to the SON SST anomalies over the tropical Indian Ocean (40°E–100°E, 5°S–5°N)[9]. The first principal pattern shows a cold-anomaly centre off Sumatra–Java, analogous to that associated with the DMI. The second principal pattern exhibits a cold-anomaly maximum in the eastern equatorial region, which extends westward and tends to be symmetric about the equator. As such, PC1 has the strongest correlation with DMI while PC2 has a positive quadratic relationship with PC1. The index for strong-pIOD events (S-index) and moderate-pIOD (M-index) can be calculated as $(PC1 + PC2)/\sqrt{2}$ and $(PC1 - PC2)/\sqrt{2}$, respectively. The moderate-pIOD is dominated by WEIO warm anomalies and strong-pIOD events are dominated by EEIO cold anomalies[9,15]. Therefore, the time evolution in the moderate-pIOD and strong-pIOD regimes largely resemble that of the SST variability over the WEIO and EEIO, respectively (Fig. 1c vs Supplementary Fig. 8a; Fig. 1e vs Supplementary Fig. 8c).

### Nonlinear advection
Both nonlinear zonal advection and nonlinear vertical advection set the strong-pIOD apart from the moderate-pIOD according to a heat budget analysis[6]. The nonlinear zonal advection term ($-U' \frac{\partial T'}{\partial x}$), which is the product of the anomalous west-minus-east SST gradient with the anomalous zonal currents, and nonlinear vertical advection term ($-W' \frac{\partial T'}{\partial z}$) which is the product of the anomalous upwelling and anomalous vertical temperature gradient, contribute to a substantial anomalous cooling of the eastern equatorial Indian Ocean during a pIOD event. As anomalous ocean currents are driven by anomalous winds, which are in turn related to the zonal gradient, the amplitude of both nonlinear terms is proportional to the square of easterly anomalies. Here we use the product of anomalous zonal SST gradient and zonal wind anomalies as surrogate for nonlinear advection term[6,9]. As a result of nonlinear advection, the cooling tendency grows exponentially, allowing a pIOD event to grow into a strong event. The strong cold anomalies are reflected in the strong thermocline feedback, as seen in Fig. 3a.

### Statistical significance test
A bootstrap method[18] is used to examine the one-standard-deviation range in associated time evolutions. For each timestep, 17 values from the 17 models are resampled randomly to construct 10,000 realisations of the multi-model ensemble mean. In this random resampling process, a model is allowed to be selected again. The standard deviation of the 10,000 interrealizations of multi-model ensemble mean is used for the uncertainty range (for example, grey shadow in Fig. 1a). The bootstrap method is also used to evaluate whether the difference in the multi-model ensemble mean over each century is significant (for example, bar to the right in Fig. 1b).

## Data availability
Data related to this paper are publicly available and can be downloaded from the following address. CMIP6 from https://esgf-node.llnl.gov/search/cmip6/ CMIP5 from https://esgf-node.llnl.gov/search/cmip5/.

## Code availability
The codes to calculate results associated with main figures in this study are available at https://doi.org/10.5281/zenodo.10949417. More information about the codes is available upon request.

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

## Acknowledgements
This work is supported by XDB40030000. The authors acknowledge the World Climate Research Programme's Working Group on Coupled Modelling, which led the design of CMIP5 and CMIP6 and coordinated the work, and we also thank individual climate modelling group for their effort in model simulations and projections. G.W. is supported by the Australian Government under the National Environmental Science Program.

## Author contributions
W.C. conceived the study and wrote the initial manuscript with A.S. and G.W.; G.W. performed data analysis and generated final figures.

## Competing interests
The authors declare no competing interests.
