## [Peer Review File · Nature Communications]

Decreased variability of the Indian Ocean Dipole after 2100 despite persistent global warmingREVIEWER COMMENTS

Reviewer #1 (Remarks to the Author):

Based on the 15 models from CMIP5/6, authors performed the analysis on the variability of IOD after 2100 under global warming and the decrease trend of IOD variability is identified. While some key points are still missing in the current research and more evidences/calculation should be added. The current version is not suitable to be accepted by this journal.

1. The Bjerknes feedback processes had been analyzed in the current research, while the negative feedback processes were missing. Please also show the negative feedback processes similar as positive Bjerknes feedback.
2. This research focus on the variability of IOD during 2100-2300. While, how can we believe the model output within this period is reliable?
3. How to define the thermocline depth in the current research?
4. What is the performance of 15 models used in this study on the simulation of IOD in the historical experiments? Whether all of these models well reproduce IOD events compared with the observation?
5. The current study use the DMI index, while whether the traditional DMI index is also suitable in the future 200-300 years? The spatial distribution of SSTA should be added to make it clear.
6. Fig. 3b, the time evolution of WEIO and EEIO SST should be introduced similar with Fig. 1.
7. What is the mechanism responsible for the reversal of thermocline depth after year 2100?

Reviewer #2 (Remarks to the Author):

Review: Decreased variability of the Indian Ocean Dipole after 2100 despite persistent global warming

Changes in austral winter and spring DMI SST variability from the 20th to the 21st century is examined in study. The authors used two generations of models that participate in Coupled Model Intercomparison Project phase 5 (CMIP5) and phase 6 (CMIP6) nine models from CMIP5 and eight models from CMIP6 forced under historical and a business-as-usual emission scenario project to the end of the 23rd century. They reported that IOD variability weakens after 2100 in a majority of models. Increasing atmosphere stability after 2100 and persistent ocean warming causes SST variability in the western and eastern poles of IOD continues to decrease, as a result decreased amplitude of moderate, strong and early-maturing positive IOD events. Analysis also result highlights a strongly nonlinear response of the IOD to long-term greenhouse warming. This study provides new insights on the variability of IOD under warming scenario after 2100.

Comments:

1. Major question is: How reliable are these projections after 2100? How about uncertainties? Projecting mean climate (annual mean) would be more valuable after 2100. Projections for a season after 2100 are questionable.
2. Mechanism proposed similar to the previous studies and what happens to IOD variability under stabilized greenhouse warming? The authors need to compare present results with stable greenhouse warming runs after 2100 in order to get more insights into IOD variability.
3. Is increasing atmosphere stability and continues ocean warming are irreversible processes? How about 2°C warming scenario after 2100?
4. It has been known that IOD variability is also depends upon the ENSO? How about ENSO and IOD association after 2100? How much is ENSO contributing to IOD variability?
5. 69: monthly SST anomalies for each year are referenced to the monthly climatology of a 10-year running average
6. 84: Fig 1. B, D, F – What is the procedure followed to calculate the variability for each model?
7. 97-98: the weakening in zonal wind variability and strengthening in the thermocline feedback due to the shoaling thermocline over the EEIO: What causes shoaling of thermocline or strengthening in the thermocline feedback when zonal wind variability is weak?
8. 108: Explain how the oceanic warming from west extending eastward arresting the shoaling EEIO thermocline?
9. I suggest the authors to look the atmospheric response to IOD variability in the warming scenario after 2100, which would be useful to understand the ocean-atmospheric response in detail.

Reviewer #2 (Remarks on code availability):

As mentioned by the authors the codes to calculate results associated with main figures in this study are available upon request

Response to Reviewer #1

Based on the 15 models from CMIP5/6, authors performed the analysis on the variability of IOD after 2100 under global warming and the decrease trend of IOD variability is identified. While some key points are still missing in the current research and more evidences/calculation should be added. The current version is not suitable to be accepted by this journal.

We thank the reviewer for the helpful comments. Please find our response below in blue.

1. The Bjerknes feedback processes had been analyzed in the current research, while the negative feedback processes were missing. Please also show the negative feedback processes similar as positive Bjerknes feedback.

Thanks for pointing this out. The major negative feedback during IOD growth and maturity phase is SST-cloud-radiation feedback (e.g., Hong et al. 2008a; Hong et al. 2008b; Cai et al. 2013). We calculated the response of surface shortwave radiation (SWR) to SST over the eastern equatorial Indian Ocean. As indicated by **Fig. A1**, there is a continuously weakening in the magnitude of SWR (downward positive) response to SST, suggesting that the negative feedback as a dampening for the SST growth is weakening. Such that the weakening of positive feedbacks including thermocline feedback and that associated with the weakening of the zonal wind variability is the primary cause for the weakening of IOD variability, which would otherwise have increased due to the weakening in this negative feedback. We have added the figure and associated discussions in the revised manuscript.

Fig. A1 | Time evolution in surface shortwave radiation (SWR) response to SST anomalies towards 2300 over the eastern equatorial Indian Ocean. It is calculated as the regression coefficient of SWR onto SST anomalies for each of the 100-year window. The SWR is positive downward. The black curve indicates the multi-model ensemble mean and the grey shadows indicate the one-standard-deviation range based on a Bootstrap test. The grey, blue, yellow, and orange dots indicate values over the 20th, the 21st, the 22nd, and the 23rd centuries.

References:

Hong, C.-C., Li, T., Ho, L. & Kug, J.-S. Asymmetry of the Indian Ocean dipole. Part I: Observational analysis. *J. Climate* 21, 4834–4848 (2008).

Hong, C.-C., Li, T. & Luo, J.-J. Asymmetry of the Indian Ocean dipole. Part II: Model diagnosis. *J. Climate* 21, 4849–4858 (2008).

Cai, W. et al. Projected response of the Indian Ocean Dipole to greenhouse warming. *Nat. Geosci.* 6, 999–1007 (2013).

2. This research focus on the variability of IOD during 2100-2300. While, how can we believe the model output within this period is reliable?

Whether model projections are reliable is a question not only to projections over 2100-2300, but also to projections towards the end of the 21st century. Several IPCC reports adopted *likelihood* as a calibrated language for describing projection probabilistic estimate based on the degree of inter-model agreement. For example, the increase in the amplitude of ENSO-associated rainfall variability with a high inter-model consensus is scored *likely* (Lee et al. 2021). This approach is indicative of the reliability of projection beyond the 21st century.

In the present study, the majority of the 17 climate models show a subsequent weakening of the IOD variability after the 21st century. Using the IPCC approach, the projected change in the IOD variability towards 2300 by the 17 climate models could be scored as *likely*.

References:

Lee, J. Y., J. Marotzke, G. Bala, L. Cao, S. Corti, J. P. Dunne, F. Engelbrecht, E. Fischer, J. C. Fyfe, C. Jones, A. Maycock, J. Mutemi, O. Ndiaye, S. Panickal, T. Zhou, 2021, Future Global Climate: Scenario-Based Projections and Near-Term Information. In: *Climate Change 2021: The Physical Science Basis. Contribution of Working Group I to the Sixth Assessment Report of the Intergovernmental Panel on Climate Change* [Masson-Delmotte, V., P. Zhai, A. Pirani, S. L. Connors, C. Péan, S. Berger, N. Caud, Y. Chen, L. Goldfarb, M. I. Gomis, M. Huang, K. Leitzell, E. Lonnoy, J. B. R. Matthews, T. K. Maycock, T. Waterfield, O. Yelekçi, R. Yu and B. Zhou (eds.)]. Cambridge University Press.

3. How to define the thermocline depth in the current research?

As mentioned in the Methods, thermocline depth is defined as the depth of maximum vertical temperature gradient, which is more appropriate in a warming climate (Vecchi and Soden 2007; Yang and Wang 2009; Ng et al. 2014).

References:

Vecchi, G. A., and B. J. Soden, 2007: Global warming and the weakening of the tropical circulation. *J. Climate*, 20, 4316–4340.

Yang, H., and F. Wang, 2009: Revisiting the thermocline depth in the equatorial Pacific. *J. Climate*, 22, 3856–3863.

Ng, B., W. Cai, and K. Walsh, 2014: Nonlinear feedbacks associated with the Indian Ocean dipole and their response to global warming in the GFDL-ESM2M coupled climate model. *J. Climate*, 27, 3904–3919

4. What is the performance of 15 models used in this study on the simulation of IOD in the historical experiments? Whether all of these models well reproduce IOD events compared with the observation?

We compared the IOD pattern and the IOD phase locking between observation and models focusing on the 1958-2022 period. The simulated IOD pattern in terms of multi-model ensemble mean is reasonable and similar to observed, with majority models agree on the seesaw feature between west and east (**Figs. A2a and A2b**). In terms of IOD phase locking, models are able to simulate the observed characteristic with IOD development in JJA and maturity into SON (**Fig. A2c**). Although the ensemble mean of SON amplitude is greater than observed, there is a lack of inter-model agreement on such bias with around half models simulating smaller amplitude than observation. As such, the 17 models simulate reasonable IOD characteristics. This is now added in the revised manuscript.

Fig. A2 | Observed and simulated IOD pattern and phase locking over 1958-2022. **a**, The observed IOD pattern which is calculated as the SST regression pattern associated with SON DMI using ORA-s5. **b**, The same as **a**, but for multi-model ensemble mean of simulated IOD pattern. Dotted area indicates where at least 90% models agree on the sign of multi-model ensemble mean. **c**, Observed and simulated IOD phase locking as indicated by solid and dashed black curves, respectively. Dashed grey curves indicate individual models.

5. The current study use the DMI index, while whether the traditional DMI index is also suitable in the future 200-300 years? The spatial distribution of SSTA should be added to make it clear.

Good point. We now calculated IOD patterns corresponding to the DMI for each of the four centuries from the 20th to the 23rd century. As indicated by **Fig. A3** below, the dipole patterns are similar across different time periods.

Fig. A3 | The simulated and projected IOD patterns from the 20th to the 23rd century. **a**, Simulated SON SST anomalies associated with one standard deviation of the DMI over the 20th century. Dotted area indicates where at least 90% models agree on the sign of multi-model ensemble mean. **b, c, d**, The same as **a**, but for projected SON SST anomalies over the 21st, the 22nd, and the 23rd century, respectively.

6. Fig. 3b, the time evolution of WEIO and EEIO SST should be introduced similar with Fig. 1.

Thanks. Please see **Fig. A4** below and the updated Fig. 3.

Fig. A4 | Time evolution of WEIO and EEIO SST variability over JJA in a 100-year sliding window from 1860 to 2300. a, JJA SST variability over the WEIO (50°E-70°E, 10°S-10°N). The black curve indicates the multi-model ensemble mean, and the grey shadows indicate the one-standard-deviation range based on a Bootstrap test (see ‘Statistical significance test’ in Methods). The red curve shows the multi-model ensemble mean of global mean temperature (GMT) in a 100-year running average, and then subtracting the mean GMT of the first 100-year window. **b,** The same as **a**, but for JJA SST variability over the EEIO (90°E-110°E, 10°S-0°).

7. What is the mechanism responsible for the reversal of thermocline depth after year 2100?

In a transient response to global warming over the 21st century, the western equatorial Indian Ocean warms faster than the eastern equatorial Indian Ocean due to the weakening of Walker Circulation, i.e., warming towards the west from the east (before 2050 in **Fig. A5a**). However, piling up of warm water in the west gradually extends eastward, i.e., warming towards the east from the west (after 2050 in **Fig. A5a**), akin to the Pacific, where the warm pool extends eastward after a period of growth. This process is further supported by the temperature tendency difference between the west and the east (**Fig. A5b**). After the 21st century, the eastern warming rate gradually increases, hampering the shoaling of the EEIO thermocline. As such, the thermocline feedback weakens thereafter. This is now added in the revised manuscript.

Fig. A5 | The reversal of warming contrast between the WEIO and the EEIO towards 2300. **a**, The time evolution of annual mean temperature averaged between 5°S and 5°N over upper 300m depth. The dotted area indicates regions where the mean climatological temperature is statistically different from the mean over the 1860-1900 above the 90% confidence level based on a student's *t*-test. **b**, The difference in the temperature tendency between the WEIO and the EEIO. The black curve indicates the multi-model ensemble mean and the grey shadows indicate the one-standard-deviation range based on a Bootstrap test (see 'Statistical significance test' in Methods).

Response to Reviewer #2

Changes in austral winter and spring DMI SST variability from the 20th to the 21st century is examined in study. The authors used two generations of models that participate in Coupled Model Intercomparison Project phase 5 (CMIP5) and phase 6 (CMIP6) nine models from CMIP5 and eight models from CMIP6 forced under historical and a business-as-usual emission scenario project to the end of the 23rd century. They reported that IOD variability weakens after 2100 in a majority of models. Increasing atmosphere stability after 2100 and persistent ocean warming causes SST variability in the western and eastern poles of IOD continues to decrease, as a result decreased amplitude of moderate, strong and early-maturing positive IOD events. Analysis also result highlights a strongly nonlinear response of the IOD to long-term greenhouse warming. This study provides new insights on the variability of IOD under warming scenario after 2100.

We thank the reviewer for the helpful and positive comments which have greatly improved the manuscript. Please find our response below in blue.

Comments:

1. Major question is: How reliable are these projections after 2100? How about uncertainties? Projecting mean climate (annual mean) would be more valuable after 2100. Projections for a season after 2100 are questionable.

Whether model projections are reliable is a question not only to projections over 2100-2300, but also to projections towards the end of the 21st century. Several IPCC reports adopted *likelihood* as a calibrated language for describing projection probabilistic estimate based on the degree of inter-model agreement. For example, the increase in the amplitude of ENSO-associated rainfall variability with a high inter-model consensus is scored *likely* (Lee et al. 2021). This approach is indicative in assessing the reliability of projection beyond the 21st century.

In the present study, the majority of the 17 climate models show a subsequent weakening of the IOD variability after the 21st century. Using the IPCC approach, the projected change in the IOD variability towards 2300 by the 17 climate models could be scored as *likely*.

We agree with you that projected annual mean state is important. We now updated all relevant figures, including Fig. 2c, Supplementary Figs. 4, 5, 10 and 11c.

References:

Lee, J. Y., J. Marotzke, G. Bala, L. Cao, S. Corti, J. P. Dunne, F. Engelbrecht, E. Fischer, J. C. Fyfe, C. Jones, A. Maycock, J. Mutemi, O. Ndiaye, S. Panickal, T. Zhou, 2021, Future Global Climate: Scenario-Based Projections and Near-Term Information. In: Climate Change 2021: The Physical Science Basis. Contribution of Working Group I to the Sixth Assessment Report of the Intergovernmental Panel on Climate Change [Masson-Delmotte, V., P. Zhai, A. Pirani, S. L. Connors, C. Péan, S. Berger, N. Caud, Y. Chen, L. Goldfarb, M. I. Gomis, M. Huang, K. Leitzell, E. Lonnoy, J. B. R. Matthews, T. K. Maycock, T. Waterfield, O. Yelekçi, R. Yu and B. Zhou (eds.)]. Cambridge University Press.

2. Mechanism proposed similar to the previous studies and what happens to IOD variability under stabilized greenhouse warming? The authors need to compare present results with stable greenhouse warming runs after 2100 in order to get more insights into IOD variability.

Thanks for pointing this out. We repeated the same analysis using all available models under the low emission scenario, that is RCP2.6/SSP1-2.6 (see Supplementary Table 1). In contrast to the continued weakening in DMI variability after about 2050 under the high CO₂ emission scenario, DMI variability is projected to stabilize thereafter under the low emission scenario, resulting from a similar behaviour in SST variability over both the west and the east (**Fig. B1**). Consistently, the time evolution of the WEIO and the EEIO SST variability follows that of zonal wind variability and the thermocline feedback, i.e., stabilised after about 2050 towards the end of the 23rd century (**Fig. B2**).

Figures and associated discussions have been added in the revised manuscript.

Fig. B1 | Time evolution of IOD variability in a 100-year sliding window from 1860 to 2300 forced under historical and RCP2.6/SSP1-2.6 emission scenarios. a, September, October, and November (SON) SST variability of the DMI, defined as the difference in SST anomalies between the WEIO (50°E-70°E, 10°S-10°N) and the EEIO (90°E-110°E, 10°S-0°). The black curve indicates the multi-model ensemble mean and the grey shadows indicate the one-standard-deviation range based on a Bootstrap test (see ‘Statistical significance test’ in Methods). The red curve shows the multi-model ensemble mean of global mean temperature (GMT) in a 100-year running average, and then subtracting the mean GMT of the first 100-year window. **b**, DMI variability over the 20th, the 21st, the 22nd, and the 23rd centuries for each model corresponding to values indicated by the grey, blue, yellow, and orange dots in **a**. The error bar with the multi-model ensemble mean is one-standard-deviation-range based on the Bootstrap. **c** and **d**, **e** and **f**, The same as **a** and **b**, but for WEIO SST variability and EEIO SST variability, respectively.

Fig. B2 | Mechanism of IOD time evolution under historical and RCP2.6/SSP1-2.6 emission scenarios. **a**, Time evolution in September, October, and November (SON) zonal wind variability towards 2300 over the central tropical Indian Ocean (5°S-5°N, 60°E-100°E). **b**, The same as **a**, but for the thermocline feedback over the EEIO. It is calculated as the regression coefficient of SST onto the thermocline anomalies for each of the 100-year window. **c**, Time evolution of climatological thermocline depth (m) along the equatorial Indian Ocean averaged over 5°S-5°N in a 100-year sliding window. The dotted area indicates regions where the climatological thermocline is statistically different from the mean thermocline over the 1860-1900 above the 90% confidence level based on a student's t-test. The time evolution of EEIO mean thermocline depth is plotted on the right side with black and red curves indicating the multi-model ensemble mean under low, and high CO₂ emission scenario, respectively.

3. Is increasing atmosphere stability and continues ocean warming are irreversible processes? How about 2°C warming scenarios after 2100?

Similar to our response to your 2nd comment, we used data forced under the RCP2.6/SSP1-2.6 to evaluate whether the atmospheric and oceanic mean state is irreversible. This scenario may also imply how the mean state will change after the global mean temperature peaks at about 2°C (red curve in **Fig. B1a**).

The initial increasing atmosphere stability (projected to be less negative) stabilises after about 2100 under RCP2.6/SSP1-2.6 (**Fig. B3**), in stark contrast to that under the high CO₂ scenario.

We also calculated the time evolution of the annual mean ocean temperature averaged over the upper 300m depth using a 100-yr sliding window (**Fig. B4a**), the result indicates that the ocean warming extends from the east to the west before 2050 due to the weakening Walker Circulation, therefore leading to a faster warming over the WEIO compared to the EEIO (**Fig. B4b**). However, the ocean warming over the entire tropical Indian Ocean stabilized thereafter (**Fig. B4a**). Thus, the increasing atmosphere stability and oceanic warming have not shown any irreversibility by 2300.

In addition, the stabilised atmospheric stability and oceanic warming over the entire tropical Indian Ocean after the 21st century underpins the stabilised zonal wind variability and thermocline feedback (**Fig. B2**).

Fig. B3 | The time evolution of multi-model ensemble mean of annual mean vertical air temperature gradient between 200hpa and 850hpa (200 mb minus 850 mb) in a 100-year running average under the low CO2 emission scenario. This is zonally averaged between 5°S and 5°N. The dotted area indicates regions where the mean vertical air temperature gradient is statistically different from the mean over the 1860-1900 above the 90% confidence level based on a student's t-test.

Fig. B4 | Oceanic warming projection along the tropical Indian Ocean towards 2300 under RCP2.6/SSP1-2.6 emission scenario. **a**, The temporal evolution of annual mean temperature averaged between 5°S and 5°N over the upper 300m depth. The dotted area indicates regions where the mean climatological temperature is statistically different from the mean over the 1860-1900 above the 90% confidence level based on a student's *t*-test. **b**, The difference in the temperature tendency between the WEIO and the EEIO. The black curve indicates the multi-model ensemble mean and the grey shadows indicate the one-standard-deviation range based on a Bootstrap test (see 'Statistical significance test' in Methods).

4. It has been known that IOD variability is also depends upon the ENSO? How about ENSO and IOD association after 2100? How much is ENSO contributing to IOD variability?

We calculated SON Nino3.4 SST index in a similar manner as we applied to the DMI SST index. We then constructed inter-model relationship in the projected changes of both DMI and Nino3.4 variability between the 21st century and the 20th century (**Fig. B5a**), implying that the projected changes in ENSO could contribute to the changes in the IOD variability at the initial stage of future projection. However, after the 21st century, the continued weakening in the DMI variability is not related to the ENSO changes (**Figs. B5b and B5c**). Thus, changes in ENSO are unlikely to systematically contribute to the projected changes in the IOD after the 21st century. This is now added to the revised manuscript.

Fig. B5 | Inter-model relationship in the projected changes between Niño3.4 SST variability and DMI SST variability. **a**, Projected changes in Niño3.4 (5° S–5° N, 170° W–120° W) SST variability against that in DMI variability from the 20th to the 21st century focusing on SON season. The correlation coefficient, slope, and p-value are indicated. **b** and **c**, The same as **a**, but for projected changes from the 21st to the 22nd century, and from the 22nd to the 23rd century, respectively. The changes are scaled by the associated projected changes in the global mean temperature.

5. 69: monthly SST anomalies for each year are referenced to the monthly climatology of a 10-year running average

Yes. We also detailed the calculation method below.

6. 84: Fig 1. B, D, F – What is the procedure followed to calculate the variability for each model?

We firstly calculated monthly SST anomalies for each year by subtracting the monthly climatology over a 10-year window that centred the year. This will exclude the possible influence from decadal variability and longer. We then calculate its variability in a 100-yr sliding window focusing on SON season. In this way, DMI SST variability for each century in each model is represented by SST variability at the year of mid-century (dots in Fig. 1a).

We have made this clearer in the Methods. Related codes are now uploaded (see code availability).

7. 97-98: the weakening in zonal wind variability and strengthening in the thermocline feedback due to the shoaling thermocline over the EEIO: What causes shoaling of thermocline or strengthening in the thermocline feedback when zonal wind variability is weak?

The change in zonal wind variability and mean state change in the zonal wind are different climate response to global warming, thus having different impacts. Arising from the increasing atmospheric stability, the weakening zonal wind variability reduces the sensitivity of zonal wind response to the zonal gradient of SST anomalies (**Fig. B6**), which is the atmospheric component of the positive Bjerknes feedback, therefore not conducive to SST variability. In contrast, there is a strengthening of climatological easterly wind due to the weakening of Walker Circulation. This shoals the eastern thermocline, which increases the sensitivity of SST response to the thermocline over the east as the subsurface is easier to influence the surface, inducing the strengthening of the thermocline feedback.

Fig. B6 | Time evolution of zonal wind response to zonal SST gradient towards 2300. This is calculated as the regression coefficient of zonal wind anomalies over the central tropical Indian Ocean (5°S-5°N, 60°E-100°E) onto the zonal SST gradient between the WEIO and the EEIO for each of the 100-year window. The black curve indicates the multi-model ensemble mean and the grey shadows indicate the one-standard-deviation range based on a bootstrap test (see 'Statistical significance test' in Methods).

8. 108: Explain how the oceanic warming from west extending eastward arresting the shoaling EEIO thermocline?

In a transient response to global warming over the 21st century, the western equatorial Indian Ocean warms faster than the eastern equatorial Indian Ocean due to the weakening of Walker Circulation, i.e., warming towards the west from the east (before 2050 in **Fig. B7a**). However, piling up of warm water in the west gradually extends eastward, i.e., warming towards the east from the west (after 2050 in **Fig. B7a**), akin to the Pacific, where the warm pool extends eastward after a period of growth. This process is further supported by the temperature tendency difference between the west and the east (**Fig. B7b**). After the 21st century, the eastern warming rate gradually increases, hampering the shoaling of the EEIO thermocline. As such, the thermocline feedback weakens thereafter. This is now added in the revised manuscript.

Fig. B7 | The reversal of warming contrast between the WEIO and the EEIO towards 2300. **a**, The time evolution of climatological temperature averaged between 5°S and 5°N over upper 300m depth. The dotted area indicates regions where the mean climatological temperature is statistically different from the mean over the 1860-1900 above the 90% confidence level based on a student's *t*-test. **b**, The difference in the temperature tendency between the WEIO and the EEIO. The black curve indicates the multi-model ensemble mean and the grey shadows indicate the one-standard-deviation range based on a Bootstrap test (see 'Statistical significance test' in Methods).

9. I suggest the authors to look the atmospheric response to IOD variability in the warming scenario after 2100, which would be useful to understand the ocean-atmospheric response in detail.

In addition to the zonal wind response to SST as the atmospheric positive feedback as shown in Fig. B6, another important atmospheric response is the SST-cloud-radiation feedback, which is the major negative feedback during IOD growth and maturity phase (e.g., Hong et al. 2008a; Hong et al. 2008b; Cai et al. 2013). We calculated the response of surface shortwave radiation (SWR) to SST over the eastern equatorial Indian Ocean (**Fig. B8**). There is a continuously weakening in the magnitude of SWR (downward positive) response to SST, suggesting that the negative feedback as a dampening for the SST growth is weakening. As such, the weakening of positive feedbacks including thermocline feedback and the zonal wind response to the SST is the primary cause for the weakening of IOD variability, which would otherwise have increased due to the weakening in this negative feedback. We have added in the revised manuscript.

Fig. B8 | Time series of surface shortwave radiation (SWR) response to SST towards 2300 over the eastern equatorial Indian Ocean. It is calculated as the regression coefficient of SWR onto SST anomalies for each of the 100-year window. The SWR is positive downward. The black curve indicates the multi-model ensemble mean and the grey shadow indicate the one-standard-deviation range based on a Bootstrap test. The grey, blue, yellow, and orange dots indicate values over the 20th, the 21st, the 22nd, and the 23rd centuries.

References:

Hong, C.-C., Li, T., Ho, L. & Kug, J.-S. Asymmetry of the Indian Ocean dipole. Part I: Observational analysis. *J. Climate* 21, 4834–4848 (2008).

Hong, C.-C., Li, T. & Luo, J.-J. Asymmetry of the Indian Ocean dipole. Part II: Model diagnosis. *J. Climate* 21, 4849–4858 (2008).

Cai, W. et al. Projected response of the Indian Ocean Dipole to greenhouse warming. *Nat. Geosci.* 6, 999–1007 (2013).

Reviewer #2 (Remarks on code availability):

As mentioned by the authors the codes to calculate results associated with main figures in this study are available upon request

Relevant codes for analysis are now uploaded (see code availability).

REVIEWERS' COMMENTS

Reviewer #1 (Remarks to the Author):

Authors have well feedbacked my concerns.

Reviewer #2 (Remarks to the Author):

Review of the paper: Decreased variability of the Indian Ocean Dipole after 2100 despite persistent global warming.

The authors have revised the manuscript satisfactorily and all the comments are addressed properly. This work has considerable impact on the understanding of Indian Ocean variability under Global warming. I have a couple of minor comments.

1. Comment: Under historical and RCP2.6/SSP1-2.6 emission scenarios, zonal wind variability and thermocline feedback stabilise after the 21st century. However, zonal wind variability seems to be slightly decreasing but not thermocline feedback. What could be the reason for this need to be explained....

2. Comment: Inter-model relationship in the projected changes between Niño3.4 SST variability and DMI SST variability. It is essential to elaborate how ENSO-IOD relationship change is calculated. Why the change in relationship is weak in the 22nd and 23rd century?

Reviewer #2 (Remarks on code availability):

Codes provided are useful.

Response to Reviewer #2

Review of the paper: Decreased variability of the Indian Ocean Dipole after 2100 despite persistent global warming.

The authors have revised the manuscript satisfactorily and all the comments are addressed properly. This work has considerable impact on the understanding of Indian Ocean variability under Global warming. I have a couple of minor comments.

We thank the reviewer again for the helpful and positive comments which have greatly improved the manuscript. Please find our response below in blue.

1. Comment: Under historical and RCP2.6/SSP1-2.6 emission scenarios, zonal wind variability and thermocline feedback stabilise after the 21st century. However, zonal wind variability seems to be slightly decreasing but not thermocline feedback. What could be the reason for this need to be explained....

Although the multi-model ensemble mean shows a slight reduction in zonal wind variability, there is no inter-model consensus with only 8 out of the 14 available models (Supplementary Table 1) showing a reduction.

2. Comment: Inter-model relationship in the projected changes between Niño3.4 SST variability and DMI SST variability. It is essential to elaborate how ENSO-IOD relationship change is calculated. Why the change in relationship is weak in the 22nd and 23rd century?

The change in ENSO-IOD relationship is calculated as the inter-model relationship between changes in ENSO and IOD variability, from the 20th to the 21st century, from the 21st to the 22nd century, and from the 22nd to the 23rd century (Supplementary Fig. 7). While there is moderate relationship between their projected change from the 20th to the 21st century, such relationship weakens towards the 23rd century. One possible reason could be due to the weakening of ENSO variability, as indicated by the x-axis in Supplementary Figs. 7b and 7c, indicating a weakening of influence from tropical Pacific.